# Vitamin D, Its Role in Recovery after Muscular Damage Following Exercise

**DOI:** 10.3390/nu13072336

**Published:** 2021-07-08

**Authors:** Alberto Caballero-García, Alfredo Córdova-Martínez, Néstor Vicente-Salar, Enrique Roche, Daniel Pérez-Valdecantos

**Affiliations:** 1Department of Anatomy and Radiology, Health Sciences Faculty, GIR: “Physical Exercise and Aging”, Campus Universitario “Los Pajaritos”, University of Valladolid, 42004 Soria, Spain; 2Department of Biochemistry, Molecular Biology and Physiology, Health Sciences Faculty, GIR: “Physical Exercise and Aging”, Campus Universitario “Los Pajaritos”, University of Valladolid, 42004 Soria, Spain; a.cordova@uva.es (A.C.-M.); danielperezvaldecantos@gmail.com (D.P.-V.); 3Department of Applied Biology-Nutrition, Institute of Bioengineering, University Miguel Hernández, 03202 Elche, Spain; nvicente@umh.es (N.V.-S.); eroche@umh.es (E.R.); 4Alicante Institute for Health and Biomedical Research (ISABIAL), 03010 Alicante, Spain; 5CIBER Fisiopatología de la Obesidad y Nutrición (CIBEROBN), Instituto de Salud Carlos III (ISCIII), 28029 Madrid, Spain

**Keywords:** exercise, immunomodulation, inflammation, vitamin D

## Abstract

Aside from its role in bone metabolism, vitamin D is a key immunomodulatory micronutrient. The active form of vitamin D (1,25(OH)D) seems to modulate the innate immune system through different mechanisms. The vitamin is involved in the differentiation of monocytes into macrophages, increasing the phagocytic and chemotactic functions of these cells. At the same time, vitamin D enables efferocytosis and prevents immunopathology. In addition, vitamin D is involved in other processes related to immune function, such as inflammation. Regarding muscle tissue, vitamin D plays an active role in muscle inflammatory response, protein synthesis, and regulation of skeletal muscle function. Two mechanisms have been proposed: A direct role of 1,25(OH)D binding to vitamin D receptors (VDRs) in muscle cells and the modulation of calcium transport in the sarcoplasmic reticulum. This second mechanism needs additional investigation. In conclusion, vitamin D seems to be effective in cases of deficiency and/or if there is a great muscular commitment, such as in high intensity exercises.

## 1. Introduction

An important area of research is centered in the adjuvant effects of different nutritional supplements that can improve muscle recovery. Some of these supplements seem to favor muscle adaptations and decrease late-onset muscle soreness (LOMS). However, the main part of this group of supplements do not exert an ergogenic effect. Instead, they can help in post-exercise recovery, favoring a subsequent optimal performance in training and/or competition. One of these supplements is vitamin D, a key modulator of inflammatory events that can help in recovery. Deficiency in vitamin D is very common in recreational and professional athletes, conditioning the adaptive response to exercise and increasing the risk of injury and stress [1,2]. Therefore, this review presents the key aspects of vitamin D related to sport performance and recovery.

## 2. Vitamin D and Muscle

Vitamin D is an essential nutrient in many aspects related to sport performance and post-exercise recovery. The most studied function of the vitamin concerns to bone metabolism and calcium homeostasis. However, vitamin D plays a key role in modulating the function of many other cell types and tissues that are instrumental in a sports context, including immune cells and skeletal muscle fibers. Due to this immunoregulatory role, vitamin D displays a significant interest as a candidate factor to reduce post-exercise muscle damage [3]. Part of this interest is due to the finding that vitamin D receptor (VDR) is present in muscular cells. In this tissue, vitamin D regulates cell proliferation and differentiation of muscle cells as well as the transport of calcium and phosphate to intracellular compartments [4]. In this context, Ahmed et al. used vitamin D supplementation to reverse myositis-myalgia in patients treated with statins and presenting vitamin D deficiency [5]. In this line, other authors have indicated that 93% of patients undergoing nonspecific musculoskeletal pain presented vitamin D deficiency [6].

Vitamin D can be considered as an essential secosteroid hormone participating in a wide variety of physiological processes [7]. For example, vitamin D is an immunomodulatory micronutrient, involved in the differentiation of monocytes into macrophages [8,9]. In this context, vitamin D prevents immunopathology increasing the phagocytic and chemotactic capacity of macrophages, promoting efferocytosis [10]. In addition, vitamin D is involved in other processes related to immune function, such as inflammation, autophagy, oxidative stress activation together with mitochondrial dysfunction and reactive oxygen species (ROS) generation and signaling. Regarding muscle tissue, vitamin D plays an active role in muscle immune modulation, muscle inflammatory response, protein synthesis, cellular growth and regulation of skeletal muscle function [11,12,13,14].

Regulation of the immune system is instrumental in prevention and therapy of inflammatory processes. Balanced modulation of cytokine production by active immune cells is an active field of research [15]. Taking into account that adaptive and innate immunity participate in inflammatory myopathies, the most commonly used treatment is aimed at suppressing or modifying the activity of immune cells. This is based on the application of corticosteroids in combination with other immunosuppressive drugs, such as steroid-saving agents [16,17].

One of the target tissues for vitamin D is skeletal muscle. In this context, there exists a link between vitamin D deficiency and myopathy. Myopathy is characterized by a degeneration of myofibers and muscle atrophy, clinically characterized by weakness, endurance reduction, persistent inflammation and infiltration of immune and inflammatory cells into skeletal muscle [14,18]. Skeletal muscle weakness usually accompanies muscle injury or damage. In addition, this is accompanied by low serum concentrations of 25-hydroxy-vitamin D (25(OH)D). Therefore, patients with osteomalacia improve skeletal muscle strength and function by increasing serum concentrations of 25(OH)D [19]. In addition, Barker et al. [20] disclosed positive correlation between serum concentrations of 25(OH)D and strength recovery after muscle injury in subjects that were not supplemented with vitamin D. In this study, the regeneration processes induced by vitamin D in skeletal muscle were attributed to augmented proliferation and diminished apoptosis of muscle cells and not to changes in the number of satellite cells or infiltrating leukocytes [21,22].

Muscle damage is evaluated by increases in circulating muscle biomarkers, such as creatine kinase (CK), alanine aminotransferase (ALT) and aspartate aminotransferase (AST), as well as by a sustained decrease in skeletal muscle strength [23]. In rats, vitamin D reduced plasmatic CK increase and modulated the pattern of plasmatic ALT and AST after intense exercise training [24]. The mechanisms proposed to explain improved recovery in maximum isometric strength with supplemented vitamin D are apoptosis inhibition and increased protein levels in the extracellular matrix [21]. In addition, supplemental vitamin D rises the expression of vitamin D receptor (VDR) in the skeletal muscle [13,25]. This increase may affect muscle regeneration and function by regulating protein synthesis [26]. In addition, Cytochrome P450 Family 27 Subfamily B Member 1 (CYP27B1) gene expression is increased. The product of this gene seems to favor the conversion of 25(OH)D to 1,25(OH)D, the active form of vitamin D [27].

Two mechanisms have been proposed to explain the role of vitamin D in the control of muscle strength. As previously mentioned, a possible explanation would involve a direct role of 1,25(OH)D binding to VDRs in muscle cells [14,28,29]. A second mechanism that needs additional investigation suggests that vitamin D modifies calcium transport into the sarcoplasmic reticulum which is instrumental in muscle contraction [30]. These findings suggest that vitamin D seems to play a significant role in muscle performance and injury prevention. On the other hand, vitamin D is also involved in the production of cytokines [31,32]. Pro-inflammatory cytokines, such as interferon-γ (IFN- γ) and interleukin-1 (IL-1), promote inflammation and the subsequent repair processes. However, their productions are usually in balance with the production of other anti-inflammatory cytokines, such as interleucine-10 (IL-10) and interleucine 13 (IL-13). Although evidence is still scarce in humans, vitamin D seems to be involved in the increased production of anti-inflammatory cytokines by immune cells [33,34].

## 3. Vitamin D and Exercise

Many studies indicate that vitamin D deficiency can affect athletic performance as a result of an inadequate recovery [35]. This seems to be connected with reduced muscle mass and strength [36]. In addition, other authors suggest that the intake of high doses of vitamin D in individuals with no deficiency could result in a negative function of the endocrine system regulated by vitamin D [37]. Therefore, the question is whether vitamin D supplements are really useful in athletes who do not have a deficiency.

In this context, optimal levels of serum 25(OH)D seem to correlate to strength and power [38], running [39] and endurance performances [40], and aerobic ability [41]. In this context, the maximum peak performance matches with the time of the season with higher serum levels of 25(OH)D in athletes [41]. In a previous report from our laboratory, vitamin D supplementation (3000 IU/day) for 8 weeks in rowers, improved the levels of 25(OH)D. Nevertheless, this daily dose over the intervention period was not enough to confirm higher group-by-time interaction of testosterone and cortisol levels [42].

However, some published results are in conflict. For example, Koundourakis et al. reported improvements in muscle strength in soccer players, determined by assessing vertical jump [41]. However, Hamilton et al., also in soccer players, reported that vitamin D supplementation was not linked to the function of the isokinetic muscles of the lower extremities in soccer players [43]. Nevertheless, soccer players displaying lower vitamin D levels significantly presented less torque in the hamstring and quadriceps muscles compared to players with high levels of vitamin D. On the other hand, Barker et al. indicate an association between vitamin D levels, muscle power and neuromuscular performance. These authors have concluded that attenuation of muscular weakness after intense exercise results from maintaining optimal serum levels of 25(OH)D [20].

In this sense, Agergaard et al. [44] assumed that since vitamin D supplementation promotes muscle mass and performance, they might expect that supplementation would improve muscular resistance and enhance exercise-induced adaptations. However, after 4 months of supplementation with vitamin D3 (1920 IU/day) in combination with a programmed resistance training, in young men, they observed a greater reduction in the expression of myostatin mRNA (negative regulator of muscle mass) and a significant change in the number of type IIa muscle fibers. However, these changes did not result in an increase in muscle strength or hypertrophy compared to non-supplemented individuals. However, when vitamin D supplementation was carried out in older adults performing resistance training, improvements in the strength/cross-sectional area were more evident compared to younger individuals [44].

On the other hand, and taking into account published data, it would be necessary to differentiate the protocols and the variations of vitamin D according to the intensity of the exercise. In this sense, Barker et al. [20] have observed rapid increases in serum 25(OH)D when intense exercise was preceded by an initial injury. These results seem to suggest that intense exercise could modulate circulating vitamin D levels. They have indicated that the cytokine and protein increase in circulation as a result of the injury, could contribute, in part, to the observed increase in 25(OH)D.

Altogether, these data suggest that vitamin D supplementation may be effective in particular conditions. Those include a deficiency in the vitamin levels or in subjects to whom the muscle is subjected to physiological wear and tear, such as sarcopenia. Therefore, and in view of the contradictory results that are published, we think that vitamin D may be effective, if there is a deficiency and/or if there is great muscular commitment, such as in high intensity exercises.

## 4. Exercise and Inflammation

Strenuous exercise produces augmentation of neutrophils and monocytes, overwhelming immunity and increasing susceptibility to infections. In this context, cytokines released into the circulation act as extracellular messengers of these processes. The most accepted hypothesis suggests that excessive exercise and inadequate recovery favors musculoskeletal trauma. This occurs together with an increased production and release to circulation of proinflammatory cytokines, including interleukin 6 (IL-6), tumor necrosis factor-α (TNF-α), and interleukin 1β (IL-1β). The interaction of these cytokines with different systems, results in a decline of performance [45].

Cytokines are a group of polypeptides or glycoproteins mainly produced by leukocytes that regulate different functions in immune cells [46,47]. They are secreted transiently during the immune response, exerting their effects by binding to specific high affinity receptors in the cytoplasmic membrane of target cells [46,47]. Once secreted, cytokines can facilitate intercellular paracrine communication in a systemic way. Cytokines can suppress their own synthesis by autocrine, paracrine or endocrine mechanisms as well as the synthesis of other cytokines and their receptors. The mechanisms include the synthesis of eicosanoids and corticosteroids, the expression of soluble receptors and the blockade of activated signal transduction pathways [48].

Functional pleiotropy and redundancy are the main characteristics of cytokines, including interleukins, interferons (IFNs), colony stimulating factors and growth factors. IL-1, IL-6 and TNF-α are present in the majority of inflammatory processes, being, therefore, considered as therapeutic intervention targets [46,47,48,49,50,51]. Inflammation is considered a cascade of cellular and molecular events leading to increases in body temperature, and capillary dilation. These responses induced by stressors are instrumental for host defense and recovery of tissue homeostasis, initiating the removal of harmful components and tissue debris [52,53]. Nevertheless, cytokine effects on target cells and systems under an immune process is variable, displaying different levels of IL-1, IL-6 and TNF-α [46,47,48,49,50,51].

Regular physical exercise has a long-term anti-inflammatory effect, once acute inflammatory actions are resolved [52,54]. However, the post-exercise pro-inflammatory processes are associated with an increased expression of pro-inflammatory cytokines. Strenuous exercise induces an increase and mobilization of neutrophils, lymphocytes and monocytes, while suppressing cellular immunity and increasing susceptibility to infections. In addition to increased cytokine production, strenuous exercise favors apoptosis and oxidative stress in many organs and tissues [46].

On the other hand, high-intensity eccentric exercises favor exercise-induced muscle damage (EIMD) [48,55,56,57,58,59]. EIMD results in elevations of inflammatory markers (i.e., C-reactive protein/CRP) and pro-inflammatory cytokines, including IL-1, IL-6 and TNF-α among others. In addition, EIMD induces the production ROS that participate as intracellular mediators in the induction of the nuclear factor-κB (NF-κB), involved the modulation of different gene programs [60,61].

IL-6 is one of the first extracellular messengers that appears in the post-exercise response. Other messengers include IL-1, TNF-α and IL-11 [62,63]. In particular, IL-1, TNF-α and IL-6 seem to participate in protein catabolism in muscle cells [64]. Acute phase response is initiated by the presence of stress that include tissue damage and inflammation as a result of intense exercise. The objectives are to stop damage progression and trigger tissue repair. IL-6 increases the production of adrenocorticotropic hormone (ACTH) of the anterior pituitary gland that induces in the adrenal cortex the synthesis of glucocorticoids [62]. At the same time, glucocorticoids enhance the biological action of IL-6 and other cytokines on hepatocytes. Macrophages undergo the inhibitory effect of glucocorticoids, modulating the production of IL-6, limiting a dangerous feedback cycle [63]. In this context, the action of pro-inflammatory and immunomodulatory cytokines could be mediated by anti-inflammatory cytokines, including IL-1 receptor antagonist (IL-1ra), IL-6 and IL-10, together with cytokine inhibitors (cortisol, prostaglandin E2 and soluble receptors against TNF-α and IL-2). It has been described that these anti-inflammatory mediators increase significantly in blood after resistance exercises [51,64,65].

On the other hand, IFN-α and IFN-β participate in the antiviral immune response. However, IFN-γ participates as an immune and inflammatory modulator. IFN-γ is produced by NK cells, being a potent macrophage activator, inducing the nonspecific cell-mediated host defense [66,67]. In this context, IFN-γ acts a pro-inflammatory cytokine, increasing the synthesis of other inflammatory cytokines such as TNF-α, regulating the expression of TNF-α receptors [68], and stimulating the expression of Nitric oxide (NO) synthase [69].

Inflammatory cytokines released by the damaged muscle after intense exercise execution act as well as extracellular messengers. In particular IL-6, IL-8 and TNF-α trigger the phagocytic activity of recruited macrophages [70]. This results in muscle debris removal and activation of tissue repair processes [71,72].

Exercise reduces oxidative stress in many organs and tissues, including blood vessels, heart, skeletal muscle, brain and liver. During exercise, increased NO production stimulates angiogenesis and vascular permeability [69,73]. ROS produced are potent oxidants that mark released muscle proteins for phagocytosis in the injured area [74]. In addition, ROS act as intracellular messengers to stimulate the expression of antioxidant enzymes, improving the antioxidant response during exercise and in pathological situations. For instance, during a treadmill sustained exercise, oxidative stress is relieved through the inactivation and decreased expression of inflammatory messengers such as IL-6, high-sensitivity CRP, TNF-α and leukocyte differentiation antigens [71].

## 5. Why Vitamin D Supplements Improve Muscular Damage Recovery?

Many studies indicate that deficient levels of vitamin D affect muscle strength and performance. This can be negative to exercise performance in the young as well as in the elderly [75,76,77,78,79,80,81,82]. On the other hand, high blood levels of vitamin D are linked to reduced injury rates and improved performance [83]. Vitamin D can decrease inflammation by inhibiting proinflammatory cytokine production such as IL-6, that promotes the conversion of monocytes to macrophages, increasing the production of additional pro-inflammatory cytokines. The result is that vitamin D reduces the production of proinflammatory cytokines, including IFN-α, IL-2, and TNF-α [56,57,84,85].

It has been well established that deficient levels of vitamin D are associated to chronic diseases and injuries in the skeletal muscle. In order to exert its effect, vitamin D binds to the intracellular receptor VDR [57,58]. VDR–vitamin D complex modulates the expression of hundreds of genes that perform essential functions [57,58,59,60,61]. VDR–vitamin D complex also regulates NO synthesis through endotelial NOS (eNOS) activity [60]. The regulated production of NO can improve endothelial function by promoting the angiogenic activities of endothelial cells [86,87,88,89]. Moreover, the absence of the VDR gene has been shown to result in a decrease in the bioavailability of L-arginine due to the increased expression of arginase-2 that competes with eNOS for L-arginine and hydrolyzes this amino acid into ornithine and urea [60].

On the other hand, increasing evidence seems to support a key role of vitamin D in modulating inflammation. An adequate level of vitamin D in the blood is key to maintaining an anti-inflammatory response [62,63]. In this context, treatment with 1,25(OH)D suppresses the expression of pro-inflammatory cytokines [64] and the VDR–vitamin D complex participates in restoring skeletal muscle function [13,17,78,90,91,92]. The proposed mechanisms to explain vitamin D action in restoring muscle strength are: (a) Directly by binding 1,25(OH)D to VDRs in muscle cells [14,28,29]; (b) Favoring calcium transport in the sarcoplasmic reticulum that improves muscle contraction. However, this last mechanism has been tested only in rat models [14]. An alternative mechanism involves endothelial cells.

Therefore, the genomic and non-genomic effects of vitamin D on the skeletal muscle, taking into account both gene transcription and changes in calcium homeostasis, demonstrate the importance of vitamin D in muscle function [92] (Figure 1). First of all, 1,25(OH)D is instrumental for the optimal balance of Ca^2+^ and phosphorus, and subsequently the preservation of skeletal muscle homeostasis [93,94]. In this sense, it has been shown that vitamin D exerts a key role in the modulation of skeletal muscle tone and contraction [95,96]. Several studies have documented a direct association of vitamin D levels with the composition of skeletal muscle fibers, as well as power, strength and physical performance. In this context, vitamin D supplements are associated with an improved muscle performance and a reduction in falls [14,18,90].

In this context, Beaudart et al. [97] have reported that low circulating levels of vitamin D are linked to impaired muscle function, a decrease in muscle strength and muscle metabolic disorders. In this line, several reports have found that low vitamin D circulating levels resulted in muscle weakness, pain, balance and fractures in the elderly [13,14]. Altogether, this evidence suggests a significant effect of vitamin D on muscle performance and injury prevention. In this regard, clinical trials have shown that vitamin D supplements resulted in improved muscle strength [98]. Other studies indicated that vitamin D supplements could modulate mitochondrial function [99]. The same authors demonstrated that vitamin D treatment improved morphology of mitochondria by regulating the synthesis of mitochondrial proteins in cultured human muscle cells, inhibiting ROS production and mitochondrial dysfunction. In addition, vitamin D deficiency increases oxidative stress induced by mitochondrial and skeletal muscle dysfunction [100].

## 6. Conclusions

Several authors have indicated that athletes have a high risk of vitamin D deficiency. In this vein, Ogan and Pritchett [101] listed the studies indicating prevalence of vitamin D deficiency (<20 ng/mL) and low levels (<32 ng/mL) in different athletic populations. However, despite the evident benefit of vitamin D in muscle function, particularly in recovery from inflammation caused by exercise, there are still few experimental studies that demonstrate an improvement in performance after vitamin D supplementation. Therefore, musculoskeletal benefits occur when deficient or insufficient circulating levels of vitamin D (20–30 ng/mL) are corrected by providing supplements. However, no improvements in muscle function and performance are observed when subjects with already normal circulating levels of vitamin D (50 ng/mL) are supplemented. Otherwise, no apparent benefits have been observed in individuals with circulating levels of vitamin D above 50 ng/mL. Therefore, it seems clear that it is crucial to preserve optimal vitamin D levels, both for maintaining basic body functions, as well as for muscle performance, recovery and anti-inflammatory control post-exercise.

However, there is still a controversy regarding the optimal levels for supplementation. Daily vitamin D requirements have been estimated between 3000 and 5000 IU (75–125 μg/day) to meet essential needs for all tissues and cells in the body [101]. However, the intakes recommended by experts might not only cover daily metabolic requirements, but also might favor the storage of vitamin D and to increase the availability. This intake strategy seems to decrease the risk of many pathologies and likely improve sport performance. Future research is needed to answer all these questions.

Nevertheless, Carlberg and Hag [102] indicated that the dose used for vitamin D supplementation is related to the “Personal Vitamin D Response Index”. In our opinion, a supplementation with vitamin D to sport practitioners requires first to monitor at long term the circulating levels of vitamin D. The dose is still a controversial issue. According to our experience with sports professionals, the daily doses recommended by the European Endocrine Society [103] maintained at long term could mimic the effects reached by a megadose supplementation at short term.

## Figures and Tables

**Figure 1 nutrients-13-02336-f001:**
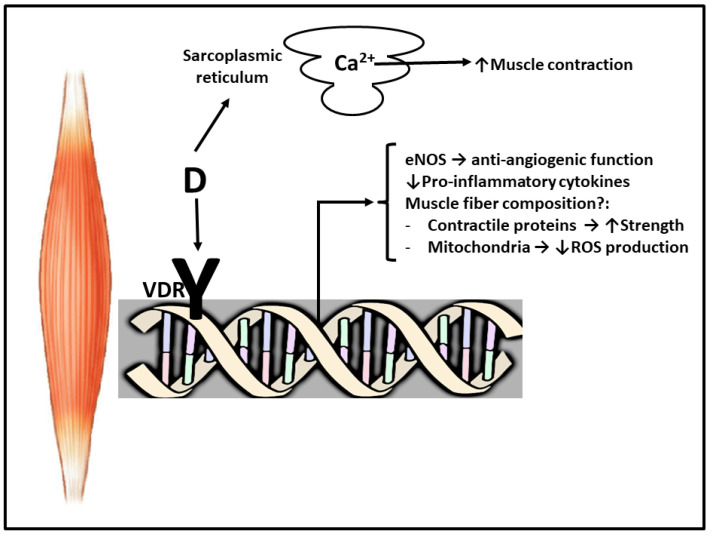
Scheme of genomic and non-genomic actions of vitamin D on skeletal muscle. Genomic actions imply binding of vitamin D (D) to the receptor (VDR). (↓) Indicates decreased expression/formation, (↑) indicates improvement/increase (?) indicates that additional research is necessary. Abbreviations used: ROS: reactive oxygen species. eNOS: endotelial NOS. See text for more details.

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
