# Peer review of "Vitamin D, Its Role in Recovery after Muscular Damage Following Exercise"

_nutrients, 2021, doi:10.3390/nu13072336_

Round 1

Reviewer 1 Report

The article is a review, presenting in a synthetic way the knowledge of the influence of vitamin D on the skeletal and muscular systems. It presents the role of this vitamin in rehabilitation and in athletes. In my opinion, the only remark concerns the number of cited works, especially in the introduction. Well-known pieces of knowledge do not require 4 or 5 references. Lines 98-99 and 108-109.

Author Response

We appreciate the suggestions proposed by the reviewer that make a more comprehensive manuscript. We have made a new distribution of citations and used less references to support certain pieces of knowledge according to Reviewer suggestions. 

Reviewer 2 Report

This is a very detailed review, referencing over 100 publications related to vitamin D influence on a wide range of endocrine and immunological activities affecting muscle cells. The authors suggest that improvements in muscle structure and function which may have been damaged by exercise, by administration of quite large intakes of vitamin D (75-125 µg/day) could be mediated either by changes in gene expression under the influence of activated vitamin D receptor in muscle cells, or by changes in calcium transport or binding in sarcoplasmic reticulum. The observation of improvements in muscle function and structure after such vitamin D treatment does suggest that there is, as the title suggests, a role for vitamin D in correcting muscle damage after exercise. However, there is a very big problem with this concept. Because functional vitamin D deficiency is very common throughout the world and particularly in athletes, the effect of dosing with vitamin D could be either (1) having a pharmacological action in muscle or (2) simply correcting a vitamin D deficiency and restoring normal vitamin D function.

A review such as this, should therefore evaluate in each of the many published studies what was the vitamin D status of the subjects before being given oral vitamin D. Was vitamin D deficiency being corrected by the treatment, or was the extra vitamin D having an effect when vitamin D status was adequate?  This distinction between correction of a deficiency and a pharmacological effect of large inputs of vitamin D is a very important requirement in a topic of the sort being considered here.

Two minor points:

  1. Line 245: “Higher serum levels of vitamin D…” does this refer to 25-hydroxyvitamin D or vitamin D itself? It is unlikely that the usually low concentration of vitamin D in serum is of much significance related to any functional change in skeletal muscle.
  2. Line 261: “..it exists increasing evidence…” This wording seems incorrect.

Author Response

This is a very interesting point raised by the Reviewer. We have taken the information of several reviews from our reference list. We have extracted the following recommendation: Muscoskeletal benefits occur when deficient or insufficient circulating levels of vitamin D (20-30ng/mL) are corrected by providing supplements. However, no improvements in muscle function and performance are observed when subjects with already normal circulating levels of vitamin D (50 ng/mL) are supplemented. Otherwise said, no apparent benefits have been observed in individuals with circulating levels of vitamin D above 50 ng/mL. This key statement is indicated in section 6 (Conclusion).

Two minor points:

  1. Line 245: “Higher serum levels of vitamin D…” does this refer to 25-hydroxyvitamin D or vitamin D itself? It is unlikely that the usually low concentration of vitamin D in serum is of much significance related to any functional change in skeletal muscle.

Answer to point 1: This sentence is supported by reference 85. This reference is a review and it refers to Vitamin D in general. Therefore the use of “vitamin D” is correct.

  1. Line 261: “..it exists increasing evidence…” This wording seems incorrect.

Answer to point 2: The sentence has been changed accordingly: “…increasing evidence seems to support…”

Round 2

Reviewer 2 Report

The point about using the term "vitamin D" for the concentration of both vitamin D and for 25-hydroxyvitamin D is that these two substances are chemically and physiologically quite different. If the term "vitamin D" is to be used for the parent molecule and its metabolites, this should be stated as: "vitamin D and its metabolites"

Author Response

The point about using the term "vitamin D" for the concentration of both vitamin D and for 25-hydroxyvitamin D is that these two substances are chemically and physiologically quite different. If the term "vitamin D" is to be used for the parent molecule and its metabolites, this should be stated as: "vitamin D and its metabolites"
